# Impact of Sea Rice Planting on Enzymatic Activity and Microbial Community of Coastal Soils: Focus on Proteinase

**Jie Yang [1], Zhiyun Liu [1], Mingyi Zhang [1], Xiaolong Zhu [1], Mingyi Wang [2], Xingfeng Xu [1,3,4] and Guangchao Liu [2,\***

1   College of Food Science and Engineering, Qingdao Agricultural University, Qingdao 266109, China; 201901168@qau.edu.cn (J.Y.); zhiyunliu2021@163.com (Z.L.); zmy020917@163.com (M.Z.); 17861422913@163.com (X.Z.); jiaonanxuxingfeng@126.com (X.X.)
2   College of Life Science, Qingdao Agricultural University, Qingdao 266109, China; 17669477903@163.com
3   Shandong Technology Innovation Center of Special Food, Qingdao 266109, China
4   Qingdao Special Food Research Institute, Qingdao 266109, China
\*   Correspondence: 201801036@qau.edu.cn

**Abstract:** Soil proteinase and proteinase-producing microbial community are closely associated with soil fertility and soil health. Sea rice has been planted in the coastal beach of Jiaozhou Bay, China, in an effort to transform saline-alkali soil into arable land. However, the knowledge regarding the bacterial degradation of organic nitrogen in sea rice soils is limited. This study aims to investigate the physicochemical characteristics and enzymatic activities of the sea rice soils, as well as the microbial communities by both the Illumina sequencing-based culture-independent technology and culture-dependent methods. Sea rice soils exhibited a lower salinity and higher organic matter content and proteinase activity, as well as an increase in both the richness and diversity of the proteinase-producing bacterial community, compared to the adjacent non-rice soils. The Proteobacteria phylum and the Gammaproteobacteria class were dominant in sea rice soils, showing higher abundance than in the reference soils. The *Planococcus* genus and *Bacillus*-like bacterial communities were abundant in the cultivable proteinase-producing bacteria isolated from sea rice soils. Furthermore, a significant proportion of the extracellular proteinase produced by the isolated soil bacteria consisted of serine proteinases and metalloproteinases. These findings provided new insights into the degradation of soil organic nitrogen in coastal agricultural regions.

**Keywords:** seawater paddy; sustainable agriculture; soil property; soil enzyme; soil microbial activity; bacterial diversity

## 1. Introduction

Coastal beach are located between marine and terrestrial ecosystem, functioning as a boundary and transitional region between them [1]. China possesses a siginificant expanse of coastal beach-lands, spanning approximately $2.17 \times 10^6$ hectares, which have facilitated the prosperous development of coastal planting, aquaculture, tourism and port industries [2]. Notably, the frequently planted species along the Chinese coastal regions include *Spartina anglica*, *S. patens*, *S. alterniflora*, *Suaeda glauca Bunge* and *Helianthus annus*. Sea rice is academically referred to as "salt-resistant rice", "salt-tolerant rice" or "salt-alkali tolerant rice". Currently, sea rice as a pioneer crop has been extensively cultivated in China's coastal agricultural fields, particularly in Jiaozhou Bay. In this region, a sea rice research and breeding center covering an area of 30 acres has been established and developed by "Father of Chinese Hybrid Rice" Longping Yuan's research team [3]. The cultivation of sea rice has the potential to enhance total grain output and aid in the saline-alkali land improvement.

Previous studies have indicated that long-term rice cultivation in mudflats leaded to a dynamic change in soil organic matter, soil microbial activity and community structure [4,5]. The application of nitrogen fertilizer to enhance crop yield resulted in a significant increase

of the total nitrogen content in the coastal planting area. Soil microflora were active in driving the biogeochemical cycling network of the rice field thereby maintaining a balance between nitrogen accumulation and metabolism [6]. Apart from inorganic nitrogen, soil contains organic nitrogen mostly as protein [7]. The studies of microbiota-associated nitrogen use in mudflat soil or saline-alkali soil mainly focused on nitrogen use efficiency [8], nitrogen fixation [6], ammonification [9], nitrification [10], and denitrification [11]. However, relatively little was unveiled about the bacterial function on organic nitrogen degradation.

Proteinases, which participate in the degradation of proteins and other nitrogen-containing organic matter, such as peptides, exert a siginificant influence on the transformation of organic nitrogen [12]. Soil proteinase activity leads to the substantial release of amino acids, and amides from soil organic nitrogen matter, which can sequently serve as substrates in the ammonification process [13]. Soil microorganisms serve sensitive indicators of soil physicochemical properties, including salinity, nutrient availability, and land use patterns [5,14]. Additionally, soil enzyme activity serves as a reliable indicator of soil microbial activity [15]. Therefore, it is necessary to explore soil proteinase-producing bacteria and soil proteinase as the driving force of nitrogen turnover in tidal field ecosystems.

The degree of amino acid sequence conservation is low among the large variety of proteinases. Up to now, the direct analysis of proteinases types in the environmental samples through the design of conserved primers for proteinase genes is nearly unattainable for molecular ecological investigations and culture-independent methods. An alternative approach involves the examination of culturable bacteria within environmental samples, whichallows for the assessment of both the diversity of the proteinase-producing bacteria and the proteinases secreted by these bacteria. Pertinent studies using culture-dependent techniques have been carried out in sediments of Jiaozhou Bay and Laizhou Bay, China, [16,17] in soils of the South Shetland Islands [18], in areas of leather tannery effluents [19] and in the surface sediment and aquaculture water of aquaculture ponds [20]. Despite the frequent isolation of proteinase-producing bacteria from paddy soils [21,22], there is a scarcity of reports regarding the diversity of such bacteria in paddy soils, particularly in the sea rice planting regions of the coastal beaches.

In this study, we collected multiple soil samples from sea rice fields and the adjacent non-rice regions on a coastal beach of Jiaozhou Bay, China. Subquently, we assessed the physicochemical properties and proteinase activity of the soil samples were, followed by the analysis of the bacterial alpha diversity and community composition using Illumina sequencing-based culture-independent technology. In addition, the richness and diversity of the proteinase-producing bacteria were investigated by culture-dependent methods. Moreover, the diversity of the bacterial extracellular proteinases was analyzed by inhibitor assay. The results of this study would shed light on the microbes-involved degradation of nitrogen-containing organic matter in coastal rice fields.

## 2. Materials and Methods

### 2.1. Sampling of Sea Rice Soils

Soil samples were collected from sea rice fields and adjacent non-rice regions in a coastal beach of Jiaozhou Bay, China. The specific locations where the samples were collected are depicted in Figure S1. Fresh soil samples were collected aseptically at depths of 2 cm to 5 cm, 5 cm to 10 cm, and 10 cm to 20 cm, respectively, which were subsequently mixed to obtain the topsoil samples (2–20 cm). Each region was sampled in triplicate. The soil samples were then sealed in sterile bags for the following physicochemical, enzymatic, and microbiological analyses.

### 2.2. Detection of Soil Physicochemical Characteristics and Proteinase Activity

The temperature was recorded in situ. The soil analysis was performed based on the experimental procedure of soil and agricultural chemistry analysis [23]. Briefly, the pH of the soil was detected at a water-soil ratio of 5:1. Total concentration of carbon, nitrogen, and phosphorus of soil samples was detected by the dichromate oxidation method,

by Kjeldahl method, and by molybdenum blue method, respectively. Soil salinity was detected by soil salts meter PNT3000 (STEPS, Hochschule, Germany). Proteinase activity of the soil samples was detected by colorimetric measurements of amino acid content as formerly described, with casein as a protein substrate [5]. The inhibitor effects on soil proteinase were detected as previously described, with proteinase inhibitors of 1.0 mM phenylmethylsulfonyl fluoride (PMSF), 1.0 mM 1,10-phenanthroline (OP), 0.1 mM trans-Epoxysuccinyl-L-leucylamido (4-guanidino) butane (E64) and 0.1 mM Pepstatin A. The chemical grade of the reagents used was analytical pure. All experiments were performed in three biological replicates, and the used instruments manufacturers' user manual and instructions were strictly followed.

### 2.3. DNA Amplification, Sequencing, and Analysis

Genomic DNA extraction of soil samples was performed using an EZNA® DNA extraction Kit (Omega, Norcross, GA, USA). V3–V4 regions of bacterial 16S rRNA gene were amplified with primers 338 F (5′-ACTCCTACGGGAGGCAGCA-3′) and 806 R (5′-GGACTACHVGGGTWTCTAAT-3′). The PCR reaction was conducted based on the protocol of the thermocycler system (GeneAmp 9700, Foster City, MA, USA). The subsequent Illumina sequencing was carried out on the sequencing platform Miseq PE300 (Majorbio Bio, Jinan, China). The analysis of the alpha-diversity of the bacterial community was performed on the Majorbio cloud platform to obtain the Shannon index, Simpson index, Sobs index, Ace index, and Chao index [24].

### 2.4. Colony Count and Isolation of Culturable Proteinase-Producing Bacteria

The sea salts were provided by Qingdao Sea-Salt Aquarium Technology, Qingdao, China. Artificial seawater was made of synthetic sea salt at a concentration of 3.5%. Culturable bacterial strains were counted and isolated using the dilution-plate method. Briefly, each soil sample were serially diluted in ten-times and spread onto the screening plates which were made of 2% gelatin, 1% casein, and 0.2% yeast extract as the protein substrates. The plates were incubated until visible colonies with transparent zone were formed, with an average time of 4 days, and then the bacterial colonies of different colors, shapes, sizes, margins and opacity were selected and sub-cultured for three times to obtain the final purified strains. The chemical grade of the used reagents was laboratory pure.

### 2.5. 16S rDNA Sequencing and Phylogenetic Analysis

Colony PCR was performed to amplify the bacterial 16S rRNA gene by using a Colony PCR kit (Mei5, Dalian, China), with primers 27F (5′-AGAGTTTGATCCTGGCTCAG-3′) and 1492R (5′-GGTTACCTTGTTACGACTTC-3′). PCR products were then sequenced using the Sanger chain termination method on the 3730 platform at Tsingke Biotechnology Co., Ltd., Beijing, China. The phylogenetic tree was carried out using the neighbor-joining statistical method, and the test of phylogeny was carried out by using the bootstrap method (MEGA version 11). The 16S rRNA gene sequences of the bacterial strains in this research were uploaded to GenBank with the accession numbers OQ618950-OQ618956, OQ618958, OQ618960-OQ618964, OQ618968-OQ618977, OQ618980-OQ618985, OQ619001-OQ619003, OQ619005, OQ619010-OQ619011, OQ619025, OQ619027-OQ619032, OQ619088, OQ619090-OQ619097, OQ619102-OQ619105, OQ619108, OQ619111, OQ619113-OQ619116, OQ619118-OQ619119, OQ619123, OQ619128-619130 and OQ651246 (Table S1).

### 2.6. Proteinase Production Ability Detection of the Isolated Strains

The proteinase-producing activity of the isolated culturable bacteria was detected using previously estabilished methods [25]. The protein substrate plates consisted of 0.2% yeast extract, and 1.5% agar in artificial seawater, with different protein substrates including milk powder (1.0%), casein, or gelatin (0.5%), respectively. The isolated culturable bacteria were streaked onto these protein plates and incubated at 25 °C for a period of three to five days. Experiments were performed in three biological replicates. The resulting plates were

then observedto measure the diameters of the hydrolytic zone (H) formed around each colony (C), and the H/C ratios were sequently calculated and recorded.

### 2.7. Inhibitor Assay on Bacterial Extracellular Proteinase

The fermentation medium utilized for the production of bacterial extracellular proteinase consisted of 0.2% yeast extract, 0.5% gelatin, 0.3% casein, and artificial seawater. Each pure bacterial strain was respectively inoculated into the fermentation medium and cultivated at 25 °C, 180 rpm for 96 h. We collected the supernatant of the fermentation medium by centrifuging for 30 min at 15,364 rcf, 4 °C, and then carried out the inhibitor assay. The supernatant was diluted to the proper concentrations with Tris-HCl buffer (50 mM, pH 8.0), and then incubated with different inhibitors, PMSF, OP, E64, and Pepstatin A, at 4 °C for one hour, respectively. The residual activity of the proteinase was detected using the previously estabilished method [25]. The sample incubated with Tris-HCl buffer in the absence of any inhibitor serves as a negative control. The inhibition ratio (%) was calculated as the the discrepancy between the relative residual activity of the proteinase sample and the control. Experiments were performed in three biological replicates.

### 2.8. Statistical Analysis

The significance test of difference in this research was assessed by paired sampling *t*-test. The level of significance was set to 5% (*p*-value < 0.05).

## 3. Results

### 3.1. Physicochemical Characteristics and Proteinase Activity of the Soil Samples

We collected soil samples from different stations of non-rice regions (Figure 1A) and sea rice fields (Figure 1B) in the coastal beach of Jiaozhou Bay. The physicochemical parameters of each sample were shown (Table 1). The sea rice and reference soils represented slightly alkaline pH of 7.98 and 8.17, respectively. The salinity of sea rice soils was 2.15, which was significantly lower compared with reference soils. The total contents of carbon, nitrogen, and phosphorus of sea rice soils were 17.48, 0.92, and 0.53 g/kg, respectively, which were higher compared with reference soils. The decrease of salinity and the increase of nitrogen, carbon, and phosphorus in sea rice soils implied that the sea rice planting improved the coastal soil as a potential agricultural field.

The enzyme activities of the soil reflect soil fertility and health. In this study, the proteinase activity of both the reference and sea rice soils was detected, revealing a significant increase in proteinase activity with sea rice cultivation (Figure 1C). Additionally, an inhibitor assay was conducted to analyze the classification of the soil proteinases. The effects of different inhibitors, including serine proteinase inhibitor PMSF, metalloproteinase inhibitor OP, aspartic proteinase inhibitor Pepstatin A, and cysteine proteinase inhibitor E64 on the proteinase activity were examined (Figure 1D). The results demonstrated that PMSF, OP, and E64 effectively inhibited the proteinase activities of the sea rice and reference soils, which indicated a majority of serine proteinase, metalloproteinase, and cysteine proteinase. However, Pepstatin A exhibited no inhibition effect on the detected soil proteinase activities, implying the extremely low content of aspartic proteinase. In addition, the inhibition effects of PMSF, OP, and E64 on the rice soil proteinase were significantly higher compared with the reference soils, which showed that serine proteinase, metalloproteinase, and cysteine proteinase occupied a higher proportion in the soils of the sea rice field compared with that of the non-rice region.

### 3.2. Alpha Diversity and Taxonomy Composition Analysis of Bacterial Community

The soil bacterial community was examined using Illumina high-throughput sequencing. The obtained effective sequence reads were applied to investigate the richness and diversity of soil microbiota. The sequencing community coverage of each sample exceeded 97%. The Alpha diversity indices were exhibited (Table 2). The Shannon index and Simpson index are the indices representing bacterial diversity. Compared with reference soils, the

diversity index of sea rice soils exhibited no significant change. The Sobs index, Ace index, and Chao index are commonly used to represent bacterial richness, all of which were significantly higher in sea rice soils than in reference soils. These results indicated that rice planting led to an increase in the bacterial richness of coastal soils.

**Table 1.** Physicochemical characteristics of sea rice soils collected from the coastal beach of Jiaozhou Bay [1].

| Station | Location | Temperature | pH | Salinity | TC | TN | TP |
|---|---|---|---|---|---|---|---|
| | (E, N) | (°C) | | (g/kg) | (g/kg) | (g/kg) | (g/kg) |
| sea rice field | 120°11′41″, 36°19′10″ | 16.8 | 7.98 ± 0.05 | 2.15 ± 0.1 * | 17.48 ± 0.62 ** | 0.92 ± 0.06 * | 0.53 ± 0.07 * |
| non-rice region | 120°11′29″, 36°18′56″ | 16.8 | 8.17 ± 0.06 | 2.6 ± 0.18 | 14.75 ± 0.34 | 0.68 ± 0.06 | 0.35 ± 0.02 |

[1] Each data is representative of at least three repeats. Statistical difference was evaluated by *t*-test. * and ** stand for significant differences between the non-rice region and the sea rice field at $p < 0.05$ and $p < 0.01$, respectively.

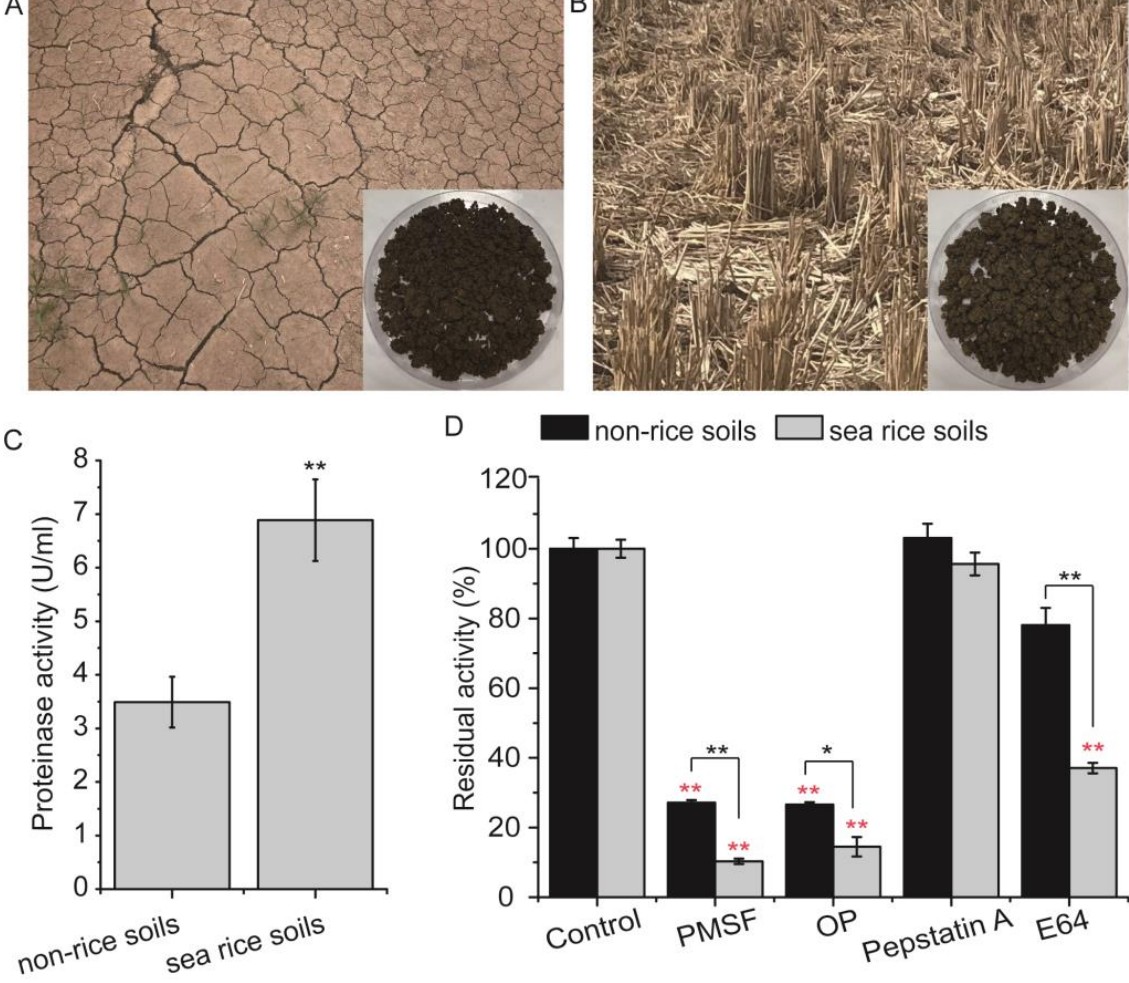

**Figure 1.** Soil sampling sites and soil proteinase activity of sea rice field and non-rice region. Fresh topsoil samples (2–20 cm) were taken for replicates from each region. (**A**) Soil sampling site and the soil of the non-rice region. (**B**) Soil sampling site and the soil of sea rice field. (**C**) Soil proteinase activity of the non-rice region and sea rice field. (**D**) Inhibition effects of PMSF, OP, Pepstatin A, and E64 on the proteinase activity of reference and sea rice soils. Each graph is representative of at least three repeats. Black *, ** means the significance of differences between the sea rice soils and reference soils at, respectively, $p < 0.05$ and $p < 0.01$; red ** means significance within the sea rice soils or reference soils in the presence of protease inhibitors at $p < 0.01$.

**Table 2.** Bacterial alpha diversity index of coastal rice soils [1].

| Sample | Coverage (%) | Diversity Index | | Richness Index | | |
|---|---|---|---|---|---|---|
| | | Shannon | Simpson $\times 10^{-3}$ | Sobs | Ace | Chao |
| sea rice soils | 98.05 ± 0.24 | 5.85 ± 0.46 | 13.90 ± 10.69 | 1789 ± 110 * | 2191 ± 119 * | 2183 ± 142 * |
| reference soils | 98.52 ± 0.12 | 5.88 ± 0.21 | 7.18 ± 2.49 | 1515 ± 70 | 1800 ± 34 | 1733 ± 31 |

[1] Each data is representative of at least three repeats. *t*-test was used to evaluate significant differences in bacterial alpha diversity between sea rice soils and reference soils. Black * means the significance of differences between the sea rice soils and reference soils at $p < 0.05$.

Furthermore, the bacterial taxonomy composition of the sea rice soils was analyzed, and the top phyla and classes were shown (Figure 2). Proteobacteria was the most dominant phylum in sea rice soils (32.10%), with a higher composition than that in reference soils (13.73%). The other major phyla in sea rice soils were Acidobacteriota (12.85), Chloroflexi (12.95%), Bacteroidota (17.63%), Actinobacteriota (5.99%), and Gemmatimonadota (8.19%) (Figure 2A). On class level, Gammaproteobacteria (23.35%), Bacteroidia (16.78%), Anaerolineae (8.69%), Vicinamibacteria (5.28%), Alphaproteobacteria (8.75%) were abundant in the sea rice soils contributing to a total of 62.85%, among which Gammaproteobacteria showed an obvious higher abundance than that in reference soils (7.86%) (Figure 2B).

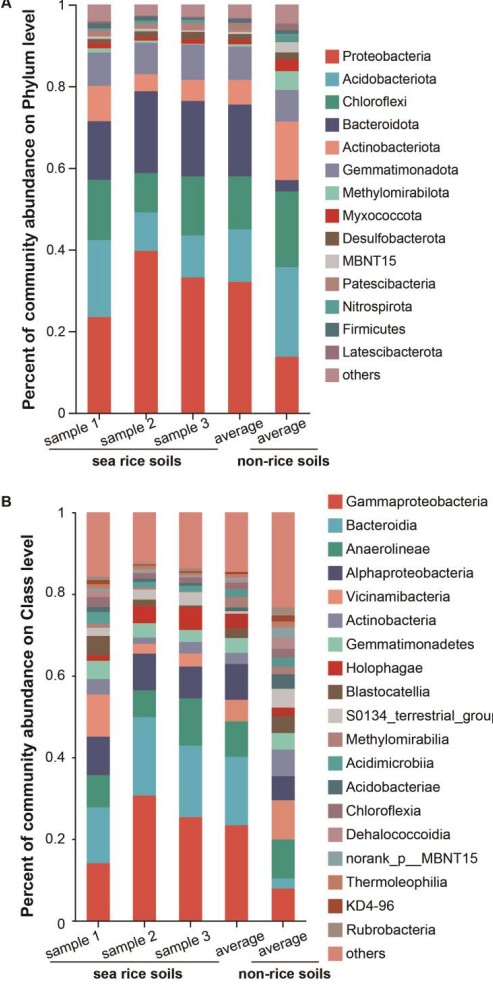

**Figure 2.** Taxonomy composition analysis of soil bacterial community. The soil samples were taken from non-rice regions and sea rice fields, and the microbial community was analyzed by using Illumina sequencing-based culture-independent technology. Composition of the bacterial community on Phylum level (**A**) and Class level (**B**). The column represents different soil samples, the row stands for the relative percentage of the microbial taxon marked in different colors.

The top 50 genera of the sea rice soil samples were shown (Figure 3). The genus with an abundance of more than 5‰ was considered the dominant genus. *Lysobacter*, *Thiobacillus*, *Antarcticibacterium*, *Sphingomonas*, *Salinimicrobium*, *Pontibacter*, *Gemmatimonas*, *Ellin6067*, *RB41*, *Gillisia*, *Zeaxanthinibacter*, *MND1*, *Erythrobacter*, and *Intrasporangium* were dominant in sea rice soils, with the sum of 34.01%. The potential proteolytic activity of the top 50 abundant genera of the bacterial community was shown (Table S2), among which *Lysobacter*, *Pontibacter*, *Arthrobacter*, *Pseudomonas*, *Bacillus*, *Streptomyces*, and *Planococcus* has been previously reported with the proteinase-producing strains [23–31], *Arthrobacter*, *Pseudomonas*, *Bacillus*, *Streptomyces*, *Planococcus*, and *Nocardioides* were isolated as proteinase-producing bacteria by the subsequent culture-dependent methods in this study.

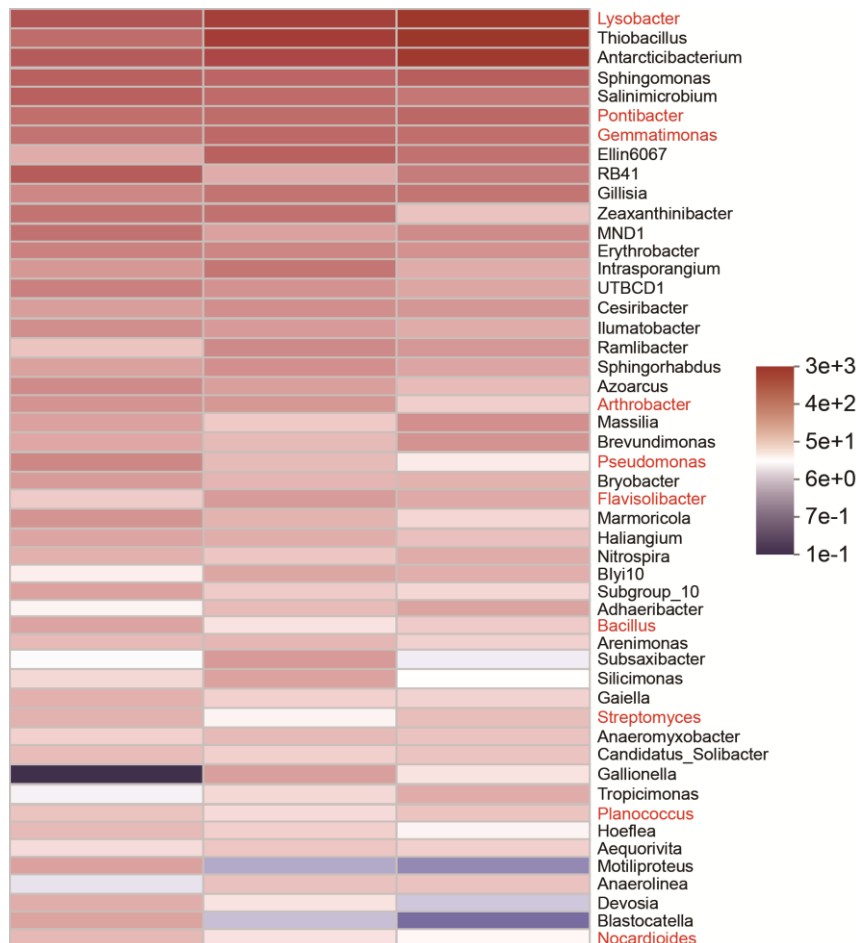

**Figure 3.** Community heatmap analysis of sea rice soils on Genus level. The heatmap was generated from the top 50 abundant OTUs according to relative abundance which was represented in different colors. The genera in red font were isolated as proteinase-producing bacteria by the subsequent culture-dependent methods or have been previously reported with the proteinase-producing strains.

### 3.3. Diversity Analysis of Proteinase-Producing Bacteria in Coastal Rice Soils

In addition to the investigation by culture-independent methods, the above soil samples of coastal rice fields and non-rice regions were furtherly used for the isolation of cultivable bacteria by culture-dependent methods. There was a large number of colonies with hydrolytic zones formed on the protein substrate plates. It was estimated by using the colony-counting method that the amount of proteinase-producing bacteria was $4.70 \pm 0.31$ Lg CFU/g soil in the sea rice field, which was significantly larger than that in non-rice soils (Figure S2). Generally, culturable microorganisms accounted for 0.1 to 1% of the total soil microbial community. The finding confirmed that a considerable sum of

proteinase-producing bacteria existed in sea rice soils, and the richness of this fraction of microbial community in sea rice soils is higher than that in non-rice soils.

The morphologically different isolates on each protein substrate plate were furtherly selected followed by repeated streaking to obtain purified cultures, and then the 16S rRNA gene of each isolate was sequenced after amplification. The GenBank accession No. of the isolated bacterial strains were shown (Table S1). In total, 68 strains of protein hydrolyzing bacteria were isolated, 26 from reference soils and 42 from sea rice soils. The neighbor-joining phylogenetic tree of these culturable proteinase-producing bacteria was constructed based on their 16S rRNA gene sequences. The phylogenetic relationship of the isolated bacterial strains from sea rice soils and reference soils was shown (Figure 4), respectively.

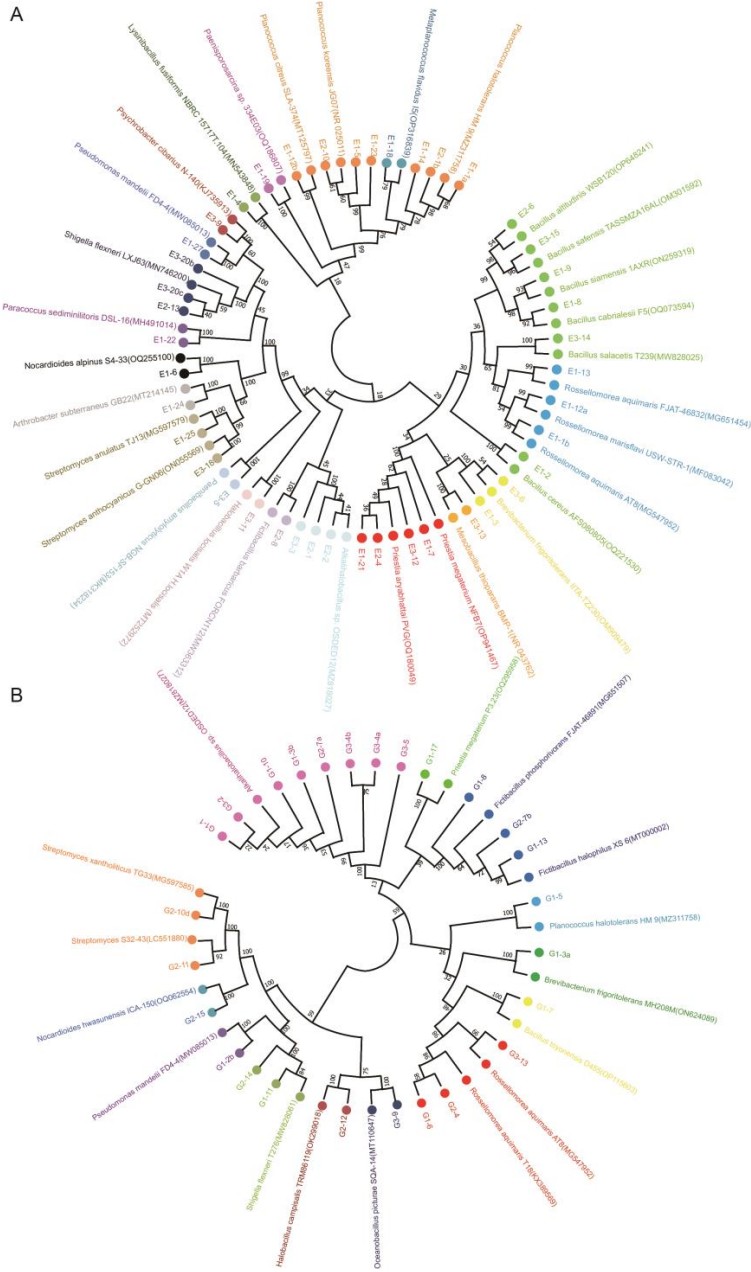

**Figure 4.** Phylogenetic relationships of the culturable proteinase-producing bacteria isolated from sea rice soils (**A**) and reference soils (**B**). Strains belonging to different genera were represented in different colors.

The protein hydrolyzing bacteria isolated from the coastal rice soils and reference soils were affiliated with 3 phyla, 4 classes, 11 families as well as 21 genera (Figure 5). These three phyla were Bacillota, Actinomycetota, and Proteobacteria. The Bacillota and Actinomycetota phyla contained only one class in the studied areas, i.e., Bacilli and Actinomycetes, respectively, while the Proteobacteria phylum contained two classes Gammaproteobacteria and Alphaproteobacteria. There were 30 proteinase-producing strains of the Bacilli class, 6 strains of the Actinomycetes class, 5 strains of the Gammaproteobacteria class, and 1 strain of the Alphaproteobacteria class in sea rice soils (Figure 5A). On the family level of cultivable proteinase-producing bacteria, Bacillaceae was found to be the largest group in the sea rice soils (45.24%), which was followed by Planococcaceae (23.81%). The Bacillaceae family was dominant in the reference soils (69.23%) (Figure 5B). On the genus level of the cultivable fraction, *Planococcus* (19.05%), *Bacillus* (11.90%) and *Priestia* (9.52%) were the major genera in the rice soils, while *Alkalihalobacillus* was abundant in the reference soils (30.77%) (Figure 5C). The other genera each contained no more than three isolates. In addition, the proteinase-producing bacterial strains cultivated from the rice soils were classified into 20 genera, while only 13 genera were from the reference soils. It seemed that the culturable proteinase-producing bacteria of rice soils were more diverse than those of reference soils.

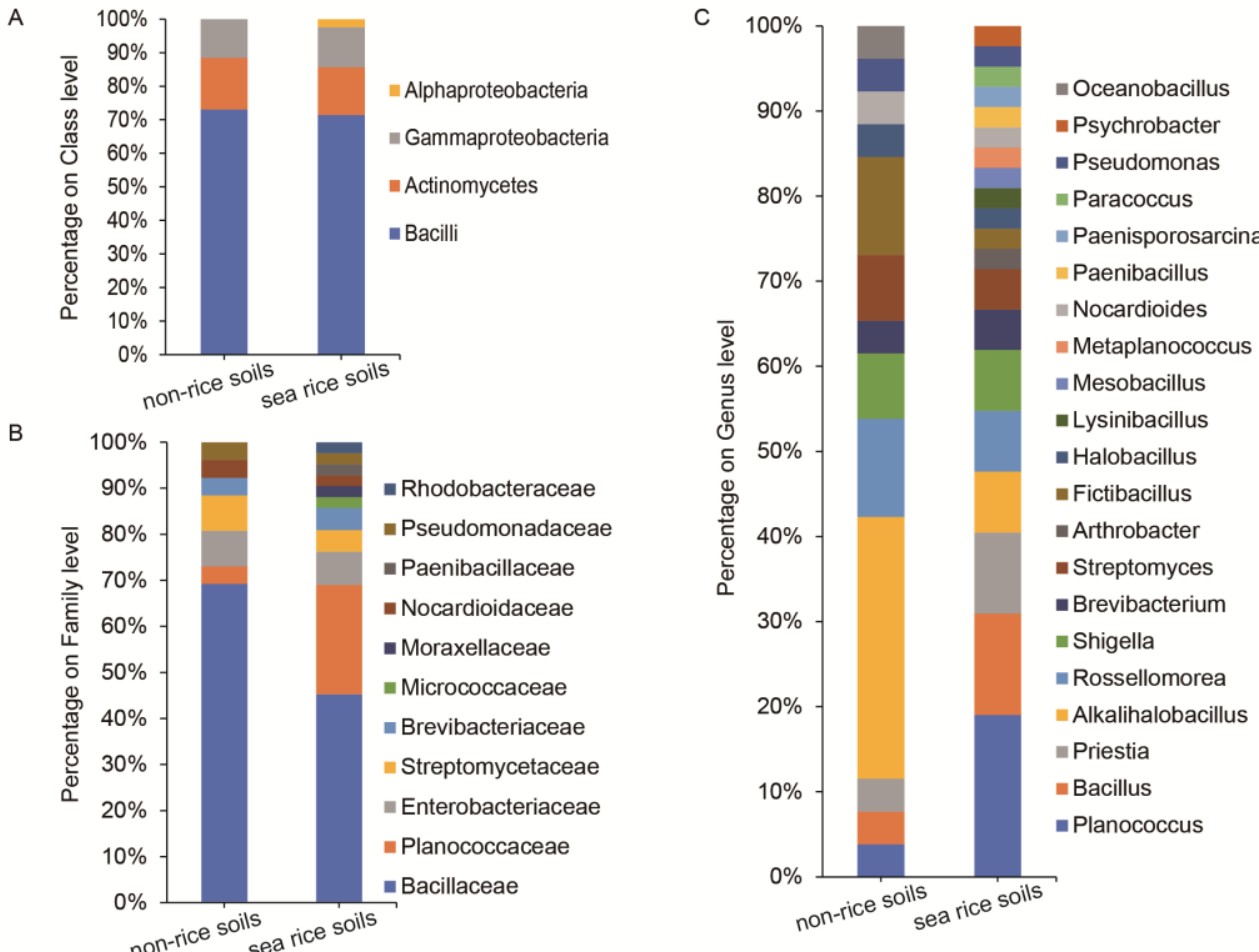

**Figure 5.** Diversity of proteinase-producing bacteria in sea rice soils. The isolates were classified on levels of Class (**A**), Family (**B**), and Genus (**C**). Different colors represented the relative percentage of each proteinase-producing taxon.

### 3.4. Diversity Analysis of the Bacterial Extracellular Proteinases

The proteinase secreted by the bacterial strains isolated from these coastal soils was furthermore investigated. The protein substrate specificity results showed that 65, 62, and 56 strains produced extracellular proteinases to hydrolyze milk powder, casein, and gelatin with clear hydrolysis circles formed on the corresponding plates, respectively (Table S3). In addition, the inhibitor assay was conducted to determine the types of extracellular proteinases. The effects of inhibitors PMSF, OP, Pepstatin A, and E64 on the proteinase activity were recorded (Table 3). A total of 68 strains were respectively cultured in the fermentation medium, and 29 strains produced enough extracellular proteinases for inhibitor assay, which were affiliated into the genera *Alkalihalobacillus*, *Bacillus*, *Fictibacillus*, *Priestia*, *Rossellomorea*, *Metaplanococcus*, *Planococcus* and *Psychrobacter*.

**Table 3.** Effects of inhibitors on the activity of bacterial extracellular proteinases.

| Genera | Strains | Inhibition Ratio (%) [1] | | | |
|---|---|---|---|---|---|
| | | PMSF [2] | OP [3] | Pepstatin A [4] | E64 [5] |
| *Alkalihalobacillus* | G2-7a | 39.11 | 45.12 | 27.77 | 10.54 |
| | E2-1 | 12.36 | 30.39 | — | — |
| | E3-3 | 17.30 | 56.89 | — | — |
| *Bacillus* | G1-7 | 40.87 | 45.77 | 20.69 | 6.44 |
| | E2-6 | 46.86 | — | — | — |
| | E1-2 | 24.86 | 26.83 | — | — |
| | E3-15 | 27.57 | 15.49 | 6.05 | — |
| | E1-9 | 48.08 | 4.24 | — | — |
| | E1-8 | 38.56 | 56.64 | 2.59 | 21.82 |
| *Fictibacillus* | G1-8 | 53.62 | — | — | — |
| | G1-13 | 34.23 | 14.92 | 11.53 | 4.29 |
| | E2-8 | 47.89 | 13.71 | — | — |
| *Lysinibacillus* | E1-4 | 34.67 | — | — | — |
| *Priestia* | G1-17 | 11.77 | 65.16 | — | — |
| | E3-12 | 14.74 | 41.86 | — | — |
| | E1-21 | 40.21 | 50.69 | 18.08 | 6.75 |
| | E1-7 | 40.65 | 42.92 | — | — |
| *Rossellomorea* | G2-4 | 41.98 | 8.21 | — | — |
| | G1-6 | 20.82 | 6.27 | — | — |
| | E1-1b | 14.17 | 28.94 | — | — |
| | E1-13 | 16.21 | 5.88 | — | — |
| *Metaplanococcus* | E1-18 | 52.16 | 21.12 | 13.44 | 9.29 |
| *Planococcus* | G1-5 | 53.81 | 17.92 | — | — |
| | G1-15 | 34.92 | 39.25 | — | — |
| | E1-23 | 39.37 | — | — | — |
| | E1-14 | 29.86 | — | — | — |
| | E2-10 | 58.30 | 28.81 | — | — |
| | E1-5 | 21.21 | 20.06 | — | — |
| *Psychrobacter* | E3-9 | 43.66 | 31.06 | 19.00 | 10.44 |

[1] The sample incubated with Tris-HCl buffer instead of any inhibitor serves as a negative control. The inhibition ratio of bacterial extracellular proteinase (%) was the difference value between the relative residual activity and the negative control. Each data is representative of at least three repeats. [2–5] The final concentration was 1.0 mM of PMSF and OP, and 0.1 mM of Pepstatin A and E64.

PMSF inhibited the proteinase ability of all 29 strains by different degrees ranging from 11.77% to 58.30%, demonstrating that all the isolated bacterial strains produced serine proteinases. OP inhibited the proteinase activity of 24 strains by 5.88%–65.16%, indicating that the majority of the bacteria secreted metalloproteinases. However, the effects of Pepstatin A and E64 on the above extracellular proteinases were not that obvious

and widespread. Pepstatin A and E64 inhibited the proteinase activity of only eight and seven strains, respectively, implying that most of the isolated strainsmay not secrete aspartic proteinase or cysteine proteinase. On the whole, the extracellular proteinases produced by the isolated coastal soil bacterial strains were mostly serine proteinases and/or metalloproteinases (Figure 6).

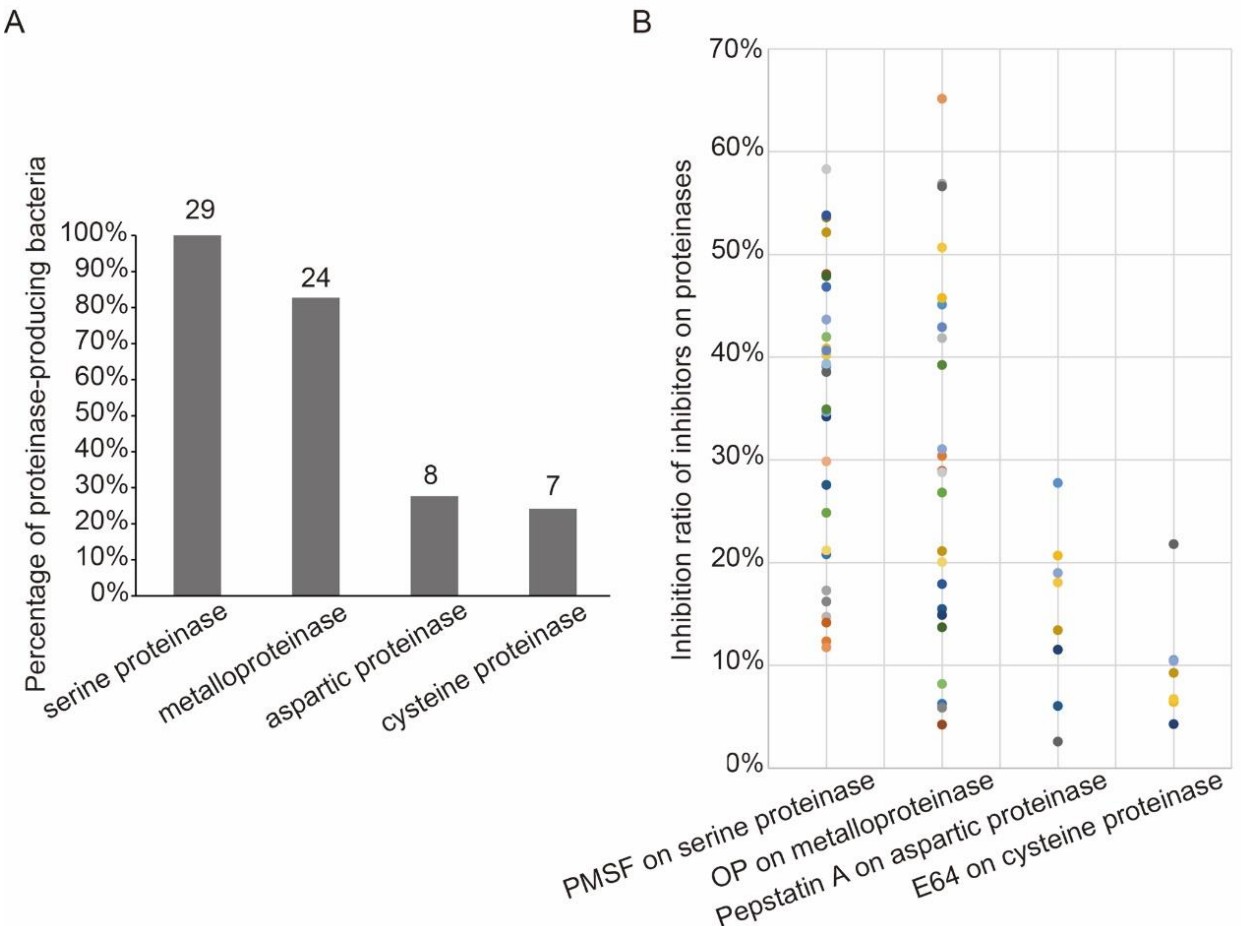

**Figure 6.** Diversity of the soil bacterial extracellular proteinases. (**A**) Percentage of proteinase-producing bacteria with different extracellular proteinases of serine proteinase, metalloproteinase, aspartic proteinase, and cysteine proteinase. The number of each type of proteinase-producing bacteria is listed above the columns. (**B**) The inhibition ratio of inhibitors PMSF, OP, Pepstatin A, and E64 on different bacterial extracellular proteinases. Different colored dots represent the extracellular proteinases produced by different bacterial strains.

## 4. Discussion

The cultivation of sea rice, also known as "salt-alkali tolerant rice", facilitated the turning of saline and alkaline land into grain fields, accounting for one-sixth of arable land in China. Soil microbes served as key players to maintain soil stability and soil health. Soil proteinase as well as proteinase-producing bacteria played an important part in the transformation of soil organic nitrogen, furtherly influencing soil fertility and plant nutrition. Accordingly, investigation of the diversity of soil proteinase and the proteinase-producing bacteria is of particular importance. In this study, culture-independent technology on the basis of Illumina sequencing was interrelated with traditional culture-dependent methods to detect the bacterial diversity in the sea rice soils of coastal beach, with an emphasis on the proteinase-producing bacteria and their extracellular proteinases.

### 4.1. The Impact of Sea Rice Planting on Physicochemical and Enzymatic Properties of Coastal Soils

We found that the total nitrogen, carbon, and phosphorus contents of sea rice soils were higher compared with that of reference soils. Previous studies also showed that long-term rice planting resulted in an enhancement of TN, TC, and TP in the rice soils [4]. Meanwhile, continuous rice cultivation led to a significant decrease in pH and salinity in the rice soil [5,32]. These findings were consistent with the physicochemical characteristics of sea rice soil we detected in this study (Table 1), indicating that rice planting was effective to improve tidal areas as new agricultural lands.

In addition to the physicochemical properties, the activity of soil enzymes, including soil proteinase, was taken as a biological indicator of soil health [5,15]. In this research, the proteinase activities of non-rice and rice soils were detected and we found a significant increase in the proteinase activity of sea rice soils compared with that of reference soils (Figure 1). Consistently in the former study, the proteinase activity in rice soil with different years of rice planting has been measured, and proteinase activity sharply increased from non-treated mudflat soil to mudflat soil after 11 years of rice cultivation, and then steadily increased from 11Y to that of 20 years' rice cultivation [5].

### 4.2. The Impact of Sea Rice Planting on the Microbial Richness and Diversity of Coastal Soils

A soil microbial community with diverse metabolic functions is crucial for crop yield and quality in agricultural lands. High salinity influenced the microbial diversity and richness in saline soils [33]. We found that the salinity of sea rice soils was lower compared with reference soils (Table 1). The decrease in salinity and the increase in organic matter showed significant effects on the bacterial community [34]. The rice soils exhibited a higher Shannon index of diversity and Chao index of richness compared with the reference soil [35]. Correspondingly in our study, the richness of bacterial community represented by the alpha indexes Sobs, Ace, and Chao, as well as the richness and diversity of proteinase-producing bacteria, increased in sea rice soils compared with that of the reference soils, indicating that the rice planting caused an enhancement of soil microbial community (Table 2).

### 4.3. The Dominant Bacterial Community and Proteinase-Producing Bacteria in Coastal Rice Soils

We investigated the bacterial taxonomy composition of the rice soils by using culture-independent technology on the basis of Illumina sequencing (Figure 2). The most dominant phylum and class were Proteobacteria and Gammaproteobacteria, which occupied 32.10% and 23.35%, respectively, in the bacterial community of sea rice soils. In addition, the Proteobacteria phylum and the Gammaproteobacteria class exhibited an increasing relative abundance to reference soils. The predominance of Proteobacteria and Gammaproteobacteria in the bacterial communities of coastal ecosystems as well as rice soils has been previously reported. The investigation of the microbial community of the sediments in Jiaozhou Bay showed that the Proteobacteria phylum was dominant accounting for 61.3%, and Gammaproteobacteria was the most abundant class with an abundance of 32.8% [36]. Proteobacteria was the most abundant bacterial phylum of the beach soils in the Huanghai Sea, and Proteobacteria exhibited an increasing relative abundance along with the rice planting years [5].

Proteinase plays an important role in the recycling of nitrogen [12,13]. To furtherly study the microbes-involved degradation of soil organic nitrogen, we investigated the proteinase-producing bacterial community in rice soils (Figure 5). A near dominance of Bacilli on the Class level was found in the soil samples. *Planococcus* in the Bacilli class was found to be the most abundant genus in rice soils with a percentage of 19.05%. It has been reported that members of the *Planococcus* genus were halotolerant and improved the salt tolerance of crops in saline soil [37]. In addition, *Bacillus*-like bacterial communities which were affiliated with ten genera in the Bacilli class, including *Bacillus*, *Priestia*, *Alkalihalobacillus*, *Rossellomorea*, *Fictibacillus*, *Halobacillus*, *Lysinibacillus*, *Mesobacillus*, *Paenibacillus* and *Oceanobacillus*, accounted for 47.62% in sea rice soils, respectively. Analogously in many other studies, *Bacillus*-like species were reported to be a dominant group in Jiaozhou Bay

sediments [17], on the coast of South India [38], in the sediments of the Hainan mangrove wetland system [35], and the continental slope of the Arabian Sea [39]. Moreover, the *Bacillus* genus was frequently reported for its proteinase-producing ability [30]. Multiple strains of the *Priestia* genus have been isolated from coastal samples [40]. The *Alkalihalobacillus* genus was reported with adaptability to high halo-alkaline [41]. *Rossellomorea* has been reported as a plant growth-promoting bacterium and a microbial inoculant to remediate heavy metal-polluted soils [42]. The plant growth-promoting property has also been found in the members of genera *Fictibacillus* [43], *Halobacillus* [44], *Lysinibacillus* [45], and *Paenibacillus* [46]. These findings demonstrated that the proteinase-producing *Planococcus* genus and *Bacillus*-like bacterial community were potential bacterial inoculants to increase the yield of the crops and improve the quality of the planting soils.

### 4.4. The Potential Application of the Soil Proteinase and Bacterial Extracellular Proteinase in the Rice Planting

The organic fertilizers applied in farming, plant debris as well as animal feces and carcasses contained a large amount of protein [47,48]. Proteinases play a major role in nitrogen mineralization in soil, by catalyzing the hydrolysis of peptide bonds of proteins to peptides and amino acids [42]. It has been reported that a fraction of organic nitrogen of small molecular weight was more easily absorbed by crops. In addition to normally used urea, amino acids and short peptides could be applied as potential nitrogen sources [49,50]. The amino acids were known as a kind of plant bio-stimulants that could promote plant growth ability and improve the plant stress resistance [51]. Amino acids showed prominent fertility performance. Exogenous amino acids enhanced the efficiency of nitrogen use, stimulated photosynthesis, and upregulated the metabolisms of nitrogen and carbon when applied to crops [51]. It has been reported that amino acids caused an upregulation of the "starch and sucrose metabolic pathway" and then promoted carbohydrate storage and energy metabolism in rice seedlings [52]. In general, peptides and amino acids, as the hydrolysis products of the in-situ protein in rice soils by proteinases, are environmentally friendly and have a certain fertility effect.

Based on the catalysis mechanism, proteinases are divided into serine proteinase, metalloproteinase, cysteine proteinase, and aspartic proteinase [53]. As for the soil proteinase detection in this study, we found the majority of sea rice soils was serine proteinase, metalloproteinase, and cysteine proteinase, with an extremely low content of aspartic proteinase. In addition, microbes that inhabit coastal sediments are key players in coastal biogeochemical cycles [54]. The proteinse-producing bacteria improved sea rice soils through the production of extracellular proteinases which degraded soil proteins. We found that serine- and metallo- proteinases were the principal types of proteinases secreted by bacterial strains in the rice soil (Figure 6). Our findings were consistent with the diversity analysis performed in sediments of the South China Sea and Jiaozhou Bay on bacterial extracellular proteinase [38,55]. To the best of our knowledge, it is the first report on the classification of the soil proteinase and the bacterial proteinase diversity of the sea rice planting soils. These findings of the soil proteinase and bacterial extracellular proteinases could be applied to develop potential proteinase preparations to improve the soil fertility in the rice fields and promote crop growth.

### 5. Conclusions

In summary, the physicochemical and enzymatic characteristics as well as the microbial community of the sea rice soils, especially the diversity of proteinase-producing bacteria, were analyzed. These results showed that sea rice planting improved the quality of the coastal soils as a result of the decrease of salinity and the increase in carbon, nitrogen, and phosphorus contents. Soil proteinase activity, as well as the richness and diversity of proteinase-producing bacteria, were also increased in the sea rice soils. Proteobacteria on the Phylum level and Gammaproteobacteria on the Class level were dominant in sea rice soils. As for proteinase-producing bacteria, the *Planococcus* genus and *Bacillus*-like

community were the most abundant groups. In addition, we found the majority of sea rice soil proteinase and bacterial extracellular proteinase were serine proteinases and metalloproteinases, which is the first report of the classification of the soil proteinase and the diversity of the bacterial proteinase of the sea rice planting soils. The transformation of the in-situ protein in coastal rice soils into peptides and amino acids by these proteinases may have a certain fertility effect and improve the coastal soil quality.

**Supplementary Materials:** The following supporting information can be downloaded at: https://www.mdpi.com/article/10.3390/agronomy13082089/s1, Figure S1: Sampling sites of the non-rice region and sea rice field on the coastal beach of Jiaozhou Bay. Figure S2: The number of proteinase-producing bacteria in sea rice soils and none-rice soils estimated by the colony-counting method. Table S1: GenBank accession No. of the isolated proteinase-producing bacterial strains. Table S2: The potential proteolytic activity of the top 50 abundant genera of the sea rice soils. Table S3: H/C ratio of the cultivable strains on protein substrate plates.

**Author Contributions:** Conceptualization, G.L.; methodology and validation, J.Y. and Z.L.; investigation, X.Z., M.W. and M.Z.; resources, G.L.; writing-original draft preparation, J.Y.; writing-review and editing, G.L.; supervision, X.X.; funding acquisition, G.L. All authors have read and agreed to the published version of the manuscript.

**Funding:** This research was funded by the National Natural Science Foundation of China, grants number 32102352, 32000004; Qingdao Agricultural University Research Fund, grants number 1120079, 1120088; the Breeding Plan of Shandong Provincial Qingchuang Research Team-Innovation Team of Functional Plant Protein-Based Food (2021); Postgraduate Innovation Program of Qingdao Agricultural University, grant number QNYCX23043; the Research Fund of Qingdao Special Food Research Institute, grant number 6602422203.

**Data Availability Statement:** The relevant data that support the results of this study are available from the first author upon request.

**Conflicts of Interest:** The authors declare no conflict of interest.

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
