# Peer review of "Impact of Sea Rice Planting on Enzymatic Activity and Microbial Community of Coastal Soils: Focus on Proteinase"

_agronomy, doi:10.3390/agronomy13082089_

Round 1

Reviewer 1 Report

The research entitled (Impact of Sea Rice Planting on Enzymatic Activity and Microbial Community of Coastal Soils: Focus on Proteinase) investigates the physicochemical characteristics, enzymatic activities and microbial communities of the sea rice. Although the research idea is good there is serious points that need to be revised throughout the manuscript. Also, the authors need to revise the material part accurately.

1.       Line 112; primers 338 F and 806 R. Cheek the primer abbreviations again and add the primer sequences.

2.       For DNA amplification, sequencing, and analysis. What was the extracted sample weight and the DNA concentration?

3.       Line 122; The plates were incubated until visible colonies with transparent zone were formed. What was the average time taken for the clear zone appearance?

4.       What were the primers used for identifying the bacterial species?

5.       Line  153; then incubated with different inhibitors at 4C for one hour, respectively. Add the inhibitor names.

6.       Although in  2.6 Proteinase production ability detection of the isolated strains; the authors detected proteinase enzyme as hallow zone diameter they put it in figure 1 as U/ml which means they detected it in liquid, not in agar plates. There is confusion in the material part, especially the enzymatic part.

7.        In Table 3 the authors put the inhibitors as abbreviations without any identification.

The research needs English editing and grammar corrections.

Author Response

The research entitled (Impact of Sea Rice Planting on Enzymatic Activity and Microbial Community of Coastal Soils: Focus on Proteinase) investigates the physicochemical characteristics, enzymatic activities and microbial communities of the sea rice. Although the research idea is good there is serious points that need to be revised throughout the manuscript. Also, the authors need to revise the material part accurately.

  1. Line 112; primers 338 F and 806 R. Cheek the primer abbreviations again and add the primer sequences.

Reply:

Thanks for your suggestions.

The primer abbreviations have been checked and the primer sequences have been added in the revised manuscript which were given as follows,

Lines 136-137:

primers 338 F (5’-ACTCCTACGGGAGGCAGCA-3’) and 806 R (5’-GGACTACHVGGGTWTCTAAT-3’)

  1. For DNA amplification, sequencing, and analysis. What was the extracted sample weight and the DNA concentration?

Reply:

Genomic DNA extraction of soil samples was performed by Majorbio Bio, China using an EZNA® DNA extraction Kit (Omega, USA). The weight of each soil sample we provided was 1.0 g, and the extracted DNA concentration was 19.20 ng/μl, 20.40 ng/μl, 21.60 ng/μl, 12.10 ng/μl, 11.80 ng/μl, respectively.

  1. Line 122; The plates were incubated until visible colonies with transparent zone were formed. What was the average time taken for the clear zone appearance?

Reply:

The earliest time taken for the clear zone appearance was 36 hours, and the average time was 4 days.

As you suggested, to revise the material part accurately, the time taken for the clear zone appearance has been added in the revised manuscript (Line 150).

  1. What were the primers used for identifying the bacterial species?

Reply:

The primers used for identifying the bacterial species were 27F (5'-AGAGTTTGATCCTGGCTCAG-3') and 1492R (5'-GGTTACCTTGTTACGACTTC-3').

As you suggested, to revise the material part accurately, the name and sequence of these primers has been added in the revised manuscript (Lines 155-156).

  1. Line 153; then incubated with different inhibitors at 4C for one hour, respectively. Add the inhibitor names.

Reply:

Thanks for your suggestion.

The inhibitors names are phenylmethylsulfonyl fluoride (PMSF), 1,10-phenanthroline (OP), E-64 and Pepstatin A, and have been added in the revised manuscript as you suggested (Line 185).

  1. Although in 2.6 Proteinase production ability detection of the isolated strains; the authors detected proteinase enzyme as hallow zone diameter they put it in figure 1 as U/ml which means they detected it in liquid, not in agar plates. There is confusion in the material part, especially the enzymatic part.

Reply:

Thanks for your comment.

Proteinase production ability detection methods in 2.6 was used to detect that of the isolated bacterial strains, which were shown in Table S3.

On the other hand, the proteinase activity in Figure 1 was that of soil samples. As you suggested, the method to detect the proteinase activity of the soil samples have been added in the revised manuscript which were given as follows,

Lines 123-126:

Proteinase activity of the soil samples was detected by colorimetric measurements of amino acid content as formerly described, with casein as a protein substrate [5].

[5] Zhang, Y.; Li, Q.; Chen, Y.; Dai, Q.; Hu, J., Dynamic change in enzyme activity and bacterial community with long-term rice cultivation in mudflats. Current Microbiology. 2019, 76, 361-369.

  1. In Table 3 the authors put the inhibitors as abbreviations without any identification.

Reply:

Thanks for your comment.

The full names of the inhibitors have been provided in the material part of the revised manuscript which were given as follows,

Lines 127-130:

with proteinase inhibitors of 1.0 mM phenylmethylsulfonyl fluoride (PMSF), 1.0 mM 1,10-phenanthroline (OP), 0.1 mM trans-Epoxysuccinyl-L-leucylamido (4-guanidino) butane (E64) and 0.1 mM Pepstatin A.

Comments on the Quality of English Language

The research needs English editing and grammar corrections.

Reply:

Thanks for your suggestion.

English editing and grammar corrections of this manuscript has been checked and revised by a colleague fluent in English writing.

Reviewer 2 Report

Journal

Agronomy (ISSN 2073-4395)

Manuscript ID

agronomy-2518057

Type

Article

Title

Impact of Sea Rice Planting on Enzymatic Activities and Microbial Community of Coastal Soils: Focus on Proteinase

Authors

Jie Yang , Zhiyun Liu , Mingyi Zhang , Xiaolong Zhu , Mingyi Wang , Xingfeng Xu , Guangchao Liu *

Topic

Soil Fertility and Plant Nutrition for Sustainable Agriculture

China is a major rice producer, accounting for about 30% of the world’s total rice production. Soil bacteria and fungi are important components of agro-ecosystems and drivers of soil nutrient cycling and sensitive indicators for assessing soil environment. The type and effectiveness of soil matrix is an important factor influencing microbial community changes including microbial community abundance, diversity and their functions.

The present manuscript topic is investigated in the literature, and there is a very few of reference published. However, this paper gives significant contribution to the current knowledge in related field. The data are sound and it deserves to be published, after minor revisions.

Overall Recommendations: Minor Revisions

Abstract:

1)      Abstract is not written as per MDPI Agronomy Journal Format.

Keywords:

2)      Keywords should not be the same as mentioned in the title or abstract.

Introduction

3)      Authors completely failed to develop the hypothesis with reference to title and objective, in the introduction section. Use www.turnitin.com to find and eliminate unnecessary self-repetition and any copied text.

Materials and methods

4)      The text has many typing and grammatical errors, capitalization issues. English style and language requires a profound revision. However, the readability of the manuscript needs to be improved, preferably carefully reviewing by a native English speaker. All proper nouns must be abbreviated. Abbreviations must be described completely at first mention with brackets. Don’t start a sentence with an abbreviation here.

5)      Please add details for analytical methodologies to make it reproducible.

6)      Quality assurance of data is mandatory!!! How many batch, repeats, chemical grade and for used instruments manufacturers’ user manual and instructions were strictly followed or not!!!

Results and discussion

7)      Data is sound one. It deserves to be published.

8)      Please cite Figure No. or Table No. in brackets at suitable places for a better connectivity in results and discussion sections as to facilitate the reader. I would have expected slightly greater discussion; more detail on the mechanisms and logical reasoning is required. There is much more scope here for discussing the implications of what the results means.

Conclusion

9)      Novelty of this research work is again questionable with reference to practical significance and economic feasibility must be worked and mentioned.

References

10)  A few very old references have been used. These must be updated with recent research findings or removed. Proper formatting is questionable. It must be according to MDPI Agriculture Journal. References formatting are inconsistent. A few DOI missing. Verify each reference from original source and cross check references in the text and reference section.

Moderate editing of English language required.

Author Response

China is a major rice producer, accounting for about 30% of the world’s total rice production. Soil bacteria and fungi are important components of agro-ecosystems and drivers of soil nutrient cycling and sensitive indicators for assessing soil environment. The type and effectiveness of soil matrix is an important factor influencing microbial community changes including microbial community abundance, diversity and their functions.

The present manuscript topic is investigated in the literature, and there is a very few of reference published. However, this paper gives significant contribution to the current knowledge in related field. The data are sound and it deserves to be published, after minor revisions.

Reply:

Thanks for your comment and recognition.

 Overall Recommendations: Minor Revisions

Abstract:

1)      Abstract is not written as per MDPI Agronomy Journal Format.

Reply:

Thanks for your suggestion.

Abstract has been written as per MDPI Agronomy Journal Format, but without headings, in the revised manuscript which were given as follows,

Lines 11-29:

1) Background: Soil proteinase and proteinase-producing microbial community are closely associated with soil fertility and soil health. Sea rice has been planted in the coastal beach of Jiaozhou Bay, China, in an effort to transform saline-alkali soil into arable land. However, the knowledge regarding the bacterial degradation of organic nitrogen in sea rice soils is limited. 2) Methods: This study aims to investigate the physicochemical characteristics and enzymatic activities of the sea rice soils, as well as the microbial communities by both the Illumina sequencing-based culture-independent technology and culture-dependent methods. 3) Results:  Sea rice soils exhibited a lower salinity and higher organic matter content and proteinase activity, as well as an increase in both the richness and diversity of the proteinase-producing bacterial community, compared to the adjacent non-rice soils. The Proteobacteria phylum and the Gammaproteobacteria class were dominant in sea rice soils, showing higher abundance than in the reference soils. The Planococcus genus and Bacillus-like bacterial communities were abundant in the cultivable proteinase-producing bacteria isolated from sea rice soils. Furthermore, a significant proportion of the extracellular proteinase produced by the isolated soil bacteria consisted of serine proteinases and metalloproteinases. 4) Conclusion: These findings provided new insights into the degradation of soil organic nitrogen in coastal agricultural regions.

Keywords:

2)      Keywords should not be the same as mentioned in the title or abstract.

Reply:

Thanks for your suggestion.

Keywords have been revised into “seawater paddy; sustainable agriculture; soil property; soil enzyme; soil microbial activity; bacterial diversity” in the revised manuscript, which are different with that mentioned in the title or abstract.

Introduction

 3)      Authors completely failed to develop the hypothesis with reference to title and objective, in the introduction section. Use www.turnitin.com to find and eliminate unnecessary self-repetition and any copied text.

Reply:

Thanks for your comment and suggestion.

Introduction has been revised in the revised manuscript which were given as follows,

Lines 36-106:

The opening paragraph of Introduction is provided with a view to the coastal beach in China and the sea rice planting in China’s coastal beach. The second paragraph is related to the influence of sea rice planting on nitrogen cycling-related soil microflora. In addition, the limit of the bacterial function on the organic nitrogen degradation is pointed out in this paragraph. The third paragraph is related to soil enzyme activity, especially proteinase, and the necessary of the study on the soil proteinase-producing bacteria and soil proteinase in tidal field ecosystem is pointed out. In the fourth paragraph, difficulty and available references of the research on the diversity of proteinase-producing bacteria and proteinase were provided. In the ending paragraph, the outline of the methods, results, conclusion of our research is provided.

In addition, the unnecessary self-repetition and copied text has been eliminated as you suggested.

Materials and methods

4)      The text has many typing and grammatical errors, capitalization issues. English style and language requires a profound revision. However, the readability of the manuscript needs to be improved, preferably carefully reviewing by a native English speaker. All proper nouns must be abbreviated. Abbreviations must be described completely at first mention with brackets. Don’t start a sentence with an abbreviation here.

Reply:

Thanks for your comment and suggestions.

English editing and grammar corrections of this manuscript has been checked and revised by a colleague fluent in English writing.

The proper nouns were abbreviated, and abbreviations were described completely at first mention with brackets, e.g., in lines 126-130:

The inhibitor effects on soil proteinase were detected as previously described, with proteinase inhibitors of 1.0 mM phenylmethylsulfonyl fluoride (PMSF), 1.0 mM 1,10-phenanthroline (OP), 0.1 mM trans-Epoxysuccinyl-L-leucylamido (4-guanidino) butane (E64) and 0.1 mM Pepstatin A.

5)      Please add details for analytical methodologies to make it reproducible.

Reply:

Thanks for your suggestion.

The material part has been revised to make it reproducible which were given in Line 107-194 in the revised manuscript.

6)      Quality assurance of data is mandatory!!! How many batch, repeats, chemical grade and for used instruments manufacturers’ user manual and instructions were strictly followed or not!!!

Reply:

To improve the quality assurance of data, the material part and the result part has been revised.

e.g., in Lines 130-132:

The chemical grade of the reagents used was analytical pure. Experiments were performed in three biological replicates, and the used instruments manufacturers’ user manual and instructions were strictly followed.

e.g., in Line 153:

The chemical grade of the used reagents was laboratory pure.

e.g., in Line 209

Each data is representative of at least three repeats.

Results and discussion

 7)      Data is sound one. It deserves to be published.

Reply:

Thanks for your comment and recognition.

8)      Please cite Figure No. or Table No. in brackets at suitable places for a better connectivity in results and discussion sections as to facilitate the reader. I would have expected slightly greater discussion; more detail on the mechanisms and logical reasoning is required. There is much more scope here for discussing the implications of what the results means.

Reply:

Thanks for your suggestion.

The Figure No. and Table No. have been cited in brackets at suitable places in results and discussion sections in the revised manuscript which were given as follows, e.g. in Line 384.

The discussion part has been revised, which were given in Line 366-479 in the revised manuscript.

Conclusion

9)      Novelty of this research work is again questionable with reference to practical significance and economic feasibility must be worked and mentioned.

Reply:

Thanks for your comment.

The practical significance and economic feasibility have not been deeply investigated, even though they might be very important.

References

10)  A few very old references have been used. These must be updated with recent research findings or removed. Proper formatting is questionable. It must be according to MDPI Agriculture Journal. References formatting are inconsistent. A few DOI missing. Verify each reference from original source and cross check references in the text and reference section.
Reply:

Thanks for your suggestion.

The old references have been updated with recent research findings or removed in the revised manuscript.

The formatting has been checked again and revised, and the missing DOI has been added.

Comments on the Quality of English Language

Moderate editing of English language required.

Reply:

Thanks for your suggestion.

English editing and grammar corrections of this manuscript has been checked and revised by a colleague fluent in English writing.

Reviewer 3 Report

My comments are in the pdf document provided.

English could be improved partially according to my instructions provided in the pdf. 

Author Response

Comments and Suggestions for Authors:

My comments are in the pdf document provided.

Comments on the Quality of English Language:

English could be improved partially according to my instructions provided in the pdf. 

Reply:

Thank you for your kindly instructions provided in the pdf document.

All the note has been accepted in the revised manuscript, e.g., in Lines 37-39, and in Line 48, and so on, too many to list out here. Thank you again.

Please explain to me what does it mean "Bunge".

Reply:

"Bunge" in Line 44,

Bunge is a kind of Suaeda glauca, which is herbs annual, to 1 m tall, light green.

I apologize for my ignorance but what is lg part of lg CFU/g, what does it mean. Please explain.

Reply:

" lg CFU/g " in Line 291,

It is the logarithm of the number of bacteria, e.g., 6 lg CFU/g= lg1000000 CFU/g.

Are rice soils fertilized in any way?

Reply:

"Discussion 4.1" in Line 381,

The data relevant to fertilization was not presented according to the references.

Round 2

Reviewer 1 Report

The authors did good work to enhance the manuscript and give sufficient answers to the questions.